Deterministic clustering based compressive sensing scheme for fog-supported heterogeneous wireless sensor networks

http://orcid.org/0000-0001-6911-4346 Osamy Walid 1 6
http://orcid.org/0000-0003-1826-6248 Aziz Ahmed 1 2 5 ahmed.aziz@fci.bu.edu.eg
http://orcid.org/0000-0001-7957-7862 M Khedr Ahmed 3 4
1 Computer Science Department, Faculty of Computers and Artificial intelligence, Benha University , Benha , Egypt
2 Tashkent State University of Economics , Tashkent , Uzbekistan
3 Computer Science Department, University of Sharjah , Sharjah , United Arab Emirate
4 Mathematic Deparment, Zagazig University , Zagazig , Egypt
5 Current affiliation: Yeoju Technical Institute in Tashkent , Tashkent , Uzbekistan
6 Department of Applied Natural Science, College of Community, Qassim University , Unaizah, Qassim , Saudi Arabia
See Chan Hwang
Electronic publication date: 2021 Apr 7
Publication date: 2021
Volume: 7
Electronic Location ID: e463
Received 2020 Oct 19; Accepted 2021 Mar 10
Copyright: © 2021 Osamy et al.
Copyright year: 2021
Copyright holder: Osamy et al.
License: This is an open access article distributed under the terms of the Creative Commons Attribution License, which permits unrestricted use, distribution, reproduction and adaptation in any medium and for any purpose provided that it is properly attributed. For attribution, the original author(s), title, publication source (PeerJ Computer Science) and either DOI or URL of the article must be cited.
License URL: https://creativecommons.org/licenses/by/4.0/

Keywords: Compressive sensing, IoT, WSNs, CS reconstruction algorithms, Fog network

Funding: The authors received no funding for this work.

==============================
Data acquisition problem in large-scale distributed Wireless Sensor Networks (WSNs) is one of the main issues that hinder the evolution of Internet of Things (IoT) technology. Recently, combination of Compressive Sensing (CS) and routing protocols has attracted much attention. An open question in this approach is how to integrate these techniques effectively for specific tasks. In this paper, we introduce an effective deterministic clustering based CS scheme (DCCS) for fog-supported heterogeneous WSNs to handle the data acquisition problem. DCCS employs the concept of fog computing, reduces total overhead and computational cost needed to self-organize sensor network by using a simple approach, and then uses CS at each sensor node to minimize the overall energy expenditure and prolong the IoT network lifetime. Additionally, the proposed scheme includes an effective algorithm for CS reconstruction called Random Selection Matching Pursuit (RSMP) to enhance the recovery process at the base station (BS) side with a complete scenario using CS. RSMP adds random selection process during the forward step to give opportunity for more columns to be selected as an estimated solution in each iteration. The results of simulation prove that the proposed technique succeeds to minimize the overall network power expenditure, prolong the network lifetime and provide better performance in CS data reconstruction.

Introduction

The Internet of Things (IoT) can be seen as the primary bridge that connects physical and digital world. IoT has become one of the significant and attractive field of research through which researchers monitor everyday usages via the Internet (Khedr et al., 2020). Integrating computational abilities in various kinds of things and living organisms can create big leap in many sectors such as health, military, home, entertainment etc (Palopoli, Passerone & Rizano, 2011). IoT consists of combinations of different technologies such as cloud computing, wireless sensor networks (WSNs), big data and data information. Nowadays, WSNs are widely used in various fields. WSNs can be considered as part of IoT, due to the huge number of connected sensor nodes it contains. The main task of IoT components (such as sensors, phones and RFID tags) is to sense, collect and store data, and then transmit the collected data to the base station (BS) (Aziz, Osamy & Khedr, 2019). However, limitation on power, computation, storage and battery resources of IoT devices hinder the development of IoT applications. To overcome these limitations most of the IoT applications depend on the cloud to deploy their computations. However, cloud solutions are unable to meet the issues such as location awareness, mobility support, geo-distribution and latency for its end users concurrently.

Fog computing can overcome these limitations of cloud computing (Palopoli, Passerone & Rizano, 2011) and bring services on the edge of the network and nearer to end users. Network devices with additional computational power and storage can be utilized as Fog servers to keep data and computation close to end users. Examples of such devices include gateways, wireless Sensors and routers (Palopoli, Passerone & Rizano, 2011; Salim, Osamy & Khedr, 2014). Fog nodes (FNs) act as middleware between cloud and the end users and offer resources to the underlying sensors. Data acquisition problem in large distributed sensor networks is one of the main challenges that hinder the further development of the IoT technology (Li, Xu & Wang, 2013).

Therefore, it is highly required to find effective techniques that solve this problem to prolong the network lifetime (Kavitha & Geetha, 2019). Various techniques have been proposed, such as routing protocols or data compression scheme (Luo, Xiang & Rosenberg, 2010; Xiang, Luo & Vasilakos, 2011). Data compression methods are used to reduce the overall data transmitted through the wireless channels, thereby reduce the energy consumed by nodes during communication.

In the perspective of data compression, compressive sensing (CS) has been regarded as a suitable technique for signal sampling and compression (Aziz, Osamy & Khedr, 2019; Mahdjane & Merazka, 2019; Arjoune et al., 2017; Osamy, Salim & Khedr, 2020).

In the context of routing algorithms, it is considered as the most important communication paradigm that can optimize energy consumption in WSNs. Designing suitable routing protocols for WSNs is a challenging issue (Choi et al., 2017; Xie & Jia, 2014). Hierarchical cluster-based routing is regarded as the most efficient protocol in terms of energy efficiency and scalability of WSNs (Aziz, Osamy & Khedr, 2019). In hierarchical protocols, sensor nodes are grouped to form clusters (Salim, Osamy & Khedr, 2014). For each cluster, one node which acts as aggregation point is called cluster head (CH) and the rest of the nodes are treated as cluster members (CMs). Each CH receives data from its CMs and the aggregated data is sent towards the BS. Finally, the BS receives these aggregated data from each CH. In this case, the total amount of transmitted data can be significantly reduced. The recent researches reveal that the integration between routing protocols and CS method can help to solve data acquisition problem (Aziz, Osamy & Khedr, 2019). However, the problem of finding an efficient way to integrate routing protocols and compressive data aggregation to decrease the data traffic is NP-complete (Aziz, Osamy & Khedr, 2019). Through this paper, we introduce an effective deterministic clustering using CS protocol (DCCS) for heterogeneous WSNs to handle the data acquisition problem. DCCS reduces the total overhead and computational cost needed to self-organize sensor network, uses CS at each sensor node to reduce the energy consumption as a whole, and increases the IoT network lifetime. In addition, we employ Fog computing infrastructure between IoT devices and the cloud for efficient saving of IoT resources. Secondly, we propose an efficient reconstruction algorithm called Random Selection Matching Pursuit (RSMP) to enhance the reconstruction operation at the BS side with a complete scenario using CS. RSMP adds random selection process during the forward step to give opportunity for more columns to be selected as an estimated solution in each iteration. The proposed scheme is validated by simulations in terms of power consumption and lifetime of the IoT network.

The paper is further structured as follows: “Literature Review” provides the literature review and in “Compressive Sensing Background”, we present a background study on CS. The newly proposed algorithms are explained in “Proposed Scheme”. The simulation and validation experiments are included in “Experiments” and finally “Conclusions” concludes our research paper.

Literature review

Many routing algorithms such as (Salim, Osamy & Khedr, 2014; Al-Zubaidi, Ariffin & Al-Qadhi, 2018; Mittal, Singh & Singh Sohi, 2017) did not take into consideration the data compression, and therefore cannot efficiently deal with the huge data traffic of WSNs. It is effective to apply compression before transmitting data to reduce total power consumption by sensor nodes. The use of CS technique can significantly reduce the total amount of data gathered and improve WSNs performance (Aziz, Osamy & Khedr, 2019; Xiang, Luo & Vasilakos, 2011; Osamy, Salim & Khedr, 2020; Haupt, Bajwa & Rabbat, 2008; Chong et al., 2009; Aziz et al., 2019).

Compressive data gathering (CDG) (Chong et al., 2009) is the primary work that used CS in WSNs. The method provides a join of CS and routing scheme for reducing the overall network energy expenditure. However, the authors of CDG did not provide analysis of their algorithm. The work in (Xiang, Luo & Vasilakos, 2011) aimed to minimize the energy expenditure by uniting compression techniques and routing schemes.

The CS scheme proposed in Haupt, Bajwa & Rabbat (2008) focused on solving the data collection problem in large-scale WSNs. In Luo et al. (2010), the authors provided a fusion of CS and tree routing methods to minimize the total forwarding energy utilization. However, it leads to an increase in the power consumed by leaf and intermediate nodes. In order to solve the tree routing issue, the authors of Luo, Xiang & Rosenberg (2010) introduced a CS strategy in a hybrid way in which only the parent nodes perform CS task. Even if this approach is convenient for small networks, cluster-based strategies turn out to be more efficient for large networks.

In Xie & Jia (2014), the authors proposed a CS hybrid method integrated with clustering and studied the connection between the cluster size and the transmissions count in hybrid CS strategy. In Salim & Osamy (2015), a multi chain based distributed routing protocol with CS (DMC-CS) is proposed. Each chain leader takes the responsibility of collecting the CS compressed samples from its CMs and then sends them to the BS. Even though this algorithm succeeds to improve the WSN lifetime, it is expensive as the BS needs to know the distances among all sensors.

EDACP-CS protocol of Khedr (2015) offers an effective technique with CS for data acquisition in heterogeneous WSN using multi-hop scheme. It integrates a cluster-based technique with CS method in which the CH selection depends on distance to BS and residual energy of nodes. However, this algorithm suffers from the computation cost overheads associated to CH selection. The work proposed in Li, Xu & Wang (2013) was the initial work which studied CS with IoT from the perspective of data-compressed sampling. The main problem of that research is that it applies CS without considering how to organize the nodes in order to transmit or receive data to and from the BS.

In the context of CS reconstruction problem, Orthogonal Matching Pursuit (OMP) (Tropp & Gilber, 2007) determines the greatest magnitude values in ΦTr index during each iteration, where, r represents the residual of y. Then, the least squares (LS) problem is solved. The works in Deanna & Roman (2009) and Donoho et al. (2012) proposed algorithms based on OMP where Stagewise Orthogonal Matching Pursuit (StOMP) proposed in Deanna & Roman (2009) is an enhancement of OMP. StOMP selects more than one column to enhance the forward step of OMP; then utilizes these columns to solve the LS problem. While in Donoho et al. (2012), OMP is enhanced by grouping the inner-products having identical magnitudes into sets; then the set with the largest energy is determined. The algorithms (Tropp & Gilber, 2007; Deanna & Roman, 2009; Donoho et al., 2012) do not have a backward step as they fall under the category of irreversible greedy algorithms. The advantage of the backward step is to recover from wrong selection that might have occurred during the forward step. On the other hand, reversible greedy algorithms e.g., IHT (Cevher & Jafarpour, 2010), CoSaMP (Needell & Tropp, 2009), SP (Wei & Olgica, 2009) and FBP (Burak & Erdogan, 2013) employ backward step to eliminate wrong selection added during the forward step.

As analyzed above, the related algorithms suffer from non-stability because they used probability-based models in each round to form the clusters. Besides, no method proposed an efficient mechanism to regularly check the suitability of selected measurement matrix in each round to decide whether to change it or not. Beside, in terms of data reconstruction algorithms mentioned above we noticed that none of them give the chance for all CS matrix columns to be tested as correct solution, which makes most of them not able to recover the original data successfully. This raises the motivation for this paper.

In this paper, we introduce an efficient CS scheme to improve the performance of WSNs, enhance the IoT network lifetime and improve the reconstruction process within a complete scenario. The proposed work consists of two algorithms:

(1) Deterministic Clustering using CS protocol (DCCS) and (2) RSMP algorithm. The highlights of our research contribution are as listed below:To overcome the stability problem and to optimize the energy consumption, DCCS algorithm distributes the network into a fixed count of clusters per round. Each cluster has a CH depending on nodes residual energy.

To overcome IoT device limitations, DCCS employs the Fog computing architecture, which is very near to the WSN nodes and hence conserves the communication energy.

In order to decrease the overall energy expenditure, DCCS divides the communication process into: (i) intra and (ii) inter cluster communication process. In intra-cluster process, DCCS organizes the nodes in each cluster into a chain using the proposed chain construction steps. In inter-cluster process, DCCS applies the same scenario used in the intra-cluster process to organize the FNs into a chain and Fog leader is selected to communicate with the cloud.

To enhance CS data gathering and reconstruction process, DCCS allows the cloud to dynamically change the measurement matrix depending on network status.

Finally, to improve the reconstruction process in the cloud side, RSMP algorithm adds a random selection process to the forward step, which give a chance for all columns to be selected as estimated solution in each round.

Cs background

The CS method allows sampling and compression to be executed in one step and this differentiates it from conventional techniques of compression where sampling and compression are performed in separate steps (Luo, Xiang & Rosenberg, 2010). In addition, the CS reconstruction strategy doesn’t require any prior knowledge to successfully recover the actual data from the compressed samples (Xiang, Luo & Vasilakos, 2011).

The general CS framework equation can be expressed as follows:

(1) y=Φx,

where, y∈Rm is the compressed samples vector, m << n, Φ is an m × n CS matrix, which is a random matrix such as Bernoulli or Gaussian distribution matrix in most of the CS methods, and signal vector x∈Rn . In this system, ||x||0 = s < m < n (Luo, Xiang & Rosenberg, 2010; Xiang, Luo & Vasilakos, 2011).

CS signal recovery

Consider the CS scenario which requires to reconstruct a larger and sparse signal using a few available measurements coefficients. One among the easiest solutions to reconstruct the signal from its available measurements using Eq. (1) is to find solution of the ‖L‖0 minimization issue that determines the count of non-zero entries and the issue of signal reconstruction becomes:

(2) x=argmin‖x‖0,Suchthat,y=Φx

Even though this works well theoretically, the problem is computationally NP-hard. It is computationally hard to determine solution to the issue defined by Eq. (2) for any vector or matrix. However, the CS framework provides efficient alternate solutions to Eq. (2) by using Basic Pursuit (Venkataramani & Bresler, 1998) or Greedy Pursuit (GP). Examples of GP includes OMP (Tropp & Gilber, 2007), StOMP (Deanna & Roman, 2009) and ROMP (Donoho et al., 2012).

Proposed scheme

Recently, IoT technologies have attracted many researchers in the area of wireless networks. However, due to the energy constraints of sensors, formulating effective data aggregation strategies and managing huge amount of information are regarded as the major challenges faced by IoT technologies. To address these problems, we introduce a new method using CS integrated with an efficient routing scheme. The proposed work consists of two algorithms:

Deterministic Clustering using CS Protocol (DCCS): during DCCS, the sensor network is converted into various clusters. A CH from each cluster is selected according to residual energy of nodes within the cluster. We assume that DCCS organizes each cluster into chain to start CS based data gathering. Moreover, it allows cloud to dynamically change the measurement matrix if it is not suitable for the network.

Random Selection Matching Pursuit (RSMP): RSMP is proposed for data reconstruction. It adds random selection during the columns selection to increase the chance of finding the correct columns in each round and improve the reconstruction performance.

In the next subsections, we describe the network assumption with the two algorithms in detail.

Network assumptions

In this work, our proposed scheme is designed according to the following assumptions:The network comprises a group of nodes with the same transmission range.

Each node belongs to one of the two classes: normal nodes, advanced nodes

Fog Node (FN) has higher level energy than normal and advanced nodes.

Finally, we use the same energy parameters as used in (Salim, Osamy & Khedr, 2014). To send a message of size l-bits to a distance d, the power consumed by the radio is:

(3) ETx(l,d)={l∗Eelec+l∗ϵfs∗d2d<d0l∗Eelec+l∗ϵmp∗d4d≥d0

In order to obtain this message, the radio expended is:

(4) ERx(l)=lEelec.

Where the radio dissipates (Eelec) = 50 nJ/bit, amplifier energy r (ϵfs) = 10 pJ/bit/m2, transmit amplifier (ϵmp)= 1310000 pJ/bit/m4, d0 = ϵfs/ϵmp. The initial energy level associated to super nodes is 2 J, for advanced nodes is 1.25 J and for normal nodes is 0.5 J.

DCCS algorithm

In DCCS algorithm, heterogeneous WSN is considered in which each of the distributed sensor nodes belong to any of the three classes: normal nodes, advanced nodes (possess relatively high energy when compared to the normal ones) and FN with a higher level of energy than normal and advanced nodes and they are positioned at predetermined locations in the WSN. DCCS succeeds to achieve a balance in total energy expenditure between nodes in every round which can lead to extension of network lifespan. The working of DCCS algorithm is presented in Fig. 1. DCCS algorithm comprises of two important phases: (1) Setup phase and (2) Data compression phase.

Figure 1 DCCS Algorithm.

Setup phase

DCCS executes this phase only once in the first round. The basic aim of this phase is to collect all sensor data X with non-CS compression in the FN with possible minimum energy consumption of sensors. To achieve that, this phase consists of four steps: CH Selection, Clusters Construction, FN Selection and Learning. We describe these steps in detail as follows:

(A) Step 1: CH Selection: This step adopts the same idea as proposed in Aderohunmu, Deng & Purvis (2011) where, the CH selection process depends only on the RE of the nodes (residual energy). In the DCCS algorithm, the Fog leader node selects a fixed number of nodes (nch) to be CHs depending on RE of every node such that the priority is for nodes with highest RE (fog leader node selection process will be described later in “Setup Phase” A). The nch value is predetermined by the cloud and can be estimated by the algorithm in Raghuvanshi et al. (2010).

After the first round, each cluster is responsible for selecting a new CH dynamically. This scenario reduces the cost for searching and selecting new CH. The selected CHs transmit their own information to all other (non-CH) nodes. The non-CH nodes will select the nearby CH to start with the Clusters Construction step.

(B) Step 2: Clusters Construction: once the selected CHs advertise themselves as CHs, the non-CH nodes start to construct clusters by selecting a closest CH Ci, where i = 1, 2, …, nch, and send join-request message (JRM) to it. This JRM contains: (1) node identification (Node-ID), the selected CH identification (CH-id), the node residual energy (Node-RE), and the node location (Node-Loc). The DCCS converts the WSN into NCH clusters where each cluster has a CH and a set of member (CM) nodes. In order to decrease the whole network power utilization for transmitting data per round in each cluster, the DCCS algorithm starts to organize the member nodes within each cluster into chain. For all clusters, each CH Ci, i = 1, 2, …, nch applies the following Initialization and Update Step to construct ChainList (s) for nodes. The procedure for this step is shown in Fig. 2, where nCMi represents the total number of members of cluster Ci, disminj=misj<k≤nCMidis (cj, ck) denotes the smallest distance between cj and other consecutive nodes ck.

Figure 2 Clusters construction algorithm.

Initialization Step: in each cluster, Ci uses its member nodes information to create the ChainListi, where ChainListi = [c0, c1, …, clast-1, clast] by adding the nearest member node c0 to it. It then updates the ChainListi with the nearest unselected member node (c1) to node c0.

Update Step: after that, Ci holds the nearest unselected neighbor node cj to node c1 in waiting to decide wherever it will be placed in ChainListi or not, by comparing the distance between c1, cj and any consecutive nodes in ChainListi. If the distance between c1 and cj is less than D, where D the distance between cj and any consecutive node, then Ci adds it to the end of ChainListi. Otherwise cj will be added between the consecutive nodes that have least distance to cj, e.g., if cr and ck are consecutive nodes in the ChainListi and if dis(cj, clast) > dis(cj, cr) and dis(cj, clast) > dis(cj, ck), then node cj will be inserted between cr and ck. Otherwise, node cj will be inserted to the end of the ChainListi after node clast, where clast is the last node of the ChainListi and dis(cj, ck) is the distance among node cj and node ck. Ci repeats the previous Update Step to include all its members in ChainListi. If a member node dies in a ChainListi, then Ci will reconstruct the chain to bypass the dead node.

By applying the previous steps, each node will send and receive with the possible minimum distance. Thus, DCCS can save power of each node.

(C) Step 3: FN Selection: each CH selects the nearest FN for transmitting its data. FN receive the data transmitted by the CHs, perform aggregation operation and deliver the data to the cloud using a routing technique.

(D) Step 4: learning process: measurement matrix selection is considered as one of the most important processes in the CS method due to its impact on nodes data where this matrix is used by sensor nodes to compress their data and is used by the cloud to reconstruct the sensor data. Incorrect selection may lead to large data loss; hence, selection of proper measurement matrix is crucial.

Each FN generates this matrix using a random seed ξ, and then broadcasts ξ to the whole WSN. For seed selection process, DCCS applies the following scenario: DCCS starts seed estimation process by learning step. During the intra-cluster process, the CH starts to collect data X by a non-CS from its chain members and then fuse these data. Then, using inter-cluster communication process, data is combined by FN towards Fog leader node and send to cloud by Fog leader node. Then, the cloud starts to find the best ξ that gives minimum error. The cloud uses this minimum error as threshold β. Finally, the cloud sends ξ to the entire network to use during Data compression phase.

Data compression phase

Deterministic Clustering based CS scheme repeats this phase starting from the second round. This phase consists of four steps: CS based data gathering within intra-cluster (cluster member to CH) and inter-cluster (FN to cloud), Reconstruction, Dynamic Re-Generation of Random Seed and CH rotation. At the end of this phase, DCCS reuses Algorithm 1 to create the cluster with the new inputs (new CHs will be the output from this phase). The details of these steps are illustrated below:

(A) CS based data gathering

As described in the previous steps, there are NCH clusters with CH Ci and chain member nodes organized in ChainListi such that each ChainListi = [c0, c1, …, clast-1, clast].

Intra-cluster (from cluster member to CH): DCCS starts CS gathering in each intra-cluster as follows: the last node clast in the ChainListi uses the global seed ξ received from the BS to generate αclast. The clast node computes its compress vector (measurement) ylast = αc_last dc_last, where dc_last is the reading of sensor clast, and then transmits the measurement yc_last to its previous neighbor node clast-1 in the ChainListi. After that, node clast-1 uses the same global seed ξ to generate αc_(last-1) and compute its measurement yc_(last-1) = αc_(last-1) dc_(last-1) and then delivers the summation vector yc_last + yc_(last-1) to the previous node clast-2. Once clast-2 receives yc_last + yc_(last-1), it computes its value yclast-2, adds it to yc_last + yc_(last-1) and then transmits the summation value to previous node in ChainListi and so on till the CH Ci. Now each CH Ci has already received the compressed vector yi = [yc_0, yc_1, …, yc_last] from their corresponding CMs. Then, each CH sends the compressed samples to the nearest FN.

Inter-cluster (from FN to cloud): Through inter-cluster communication, DCCS applies the same scenario used in Algorithm 1 to organize the FN in chain and consider them as CMs of a cluster with the cloud as CH.

The communication among FN is restricted to adjacent FNs in each round, and the fog leader node will be selected to deliver the gathered data to the cloud. As the scenario is same as in Algorithm 1, FNs are organized into a chain to deliver the information among FNs and to deliver the aggregated data to the cloud. The formed chain allows the nodes to interact and exchange information with their chain neighbors. CHs deliver their collected data to FNs, and the data will be fused at the FNs.

Finally, fog leader node will deliver the fused data to the cloud. The selection of fog leader node depends on their energy and the distance to the cloud. Formation of a chain among FNs and choosing one among them as fog leader to send the data to the cloud could save more energy in the network than sending the data independently by each FN to the cloud. The communication process for both inter-cluster and intra-cluster are shown in Fig. 3.

Figure 3 Inter and Intra cluster communication process.

(B) Reconstruction Step When the cloud gets the compress vector y = [y1, y2, y3, …,yi], where i = [1, 2, …, nch] transmitted by the FN leader, cloud generates the CS matrix depending on the predefined random seed ξ. After that, the cloud reconstructs the original data x0 of every cluster. In order to improve this step, in this paper, RSMP is proposed. The working of RSMP will be described in “RSMP Algorithm”.

(C) Dynamic re-generation of random seed

The main idea of this step is that DCCS gives the ability to dynamically change the CS matrix depending on the network status, and the number of nodes that are still alive, instead of using the same CS matrix generated during the Setup phase in all rounds.

The problem of using the fixed CS matrix is that: in each round, every sensor node transmits and receives fixed size vector whatever the count of alive nodes in each round (which should be varied according to the count of alive nodes); this leads to an increment in the average power consumption and also has negative reflection in the reconstruction process.

To overcome this problem, DCCS dynamically changes the CS matrix whenever the network status changes, i.e., the CS matrix size reduces in accordance with the number of alive nodes. In this situation, DCCS can successfully decrease the overall power consumption. The cloud can obtain the dead nodes count in every cluster from FN through the CHs. Where, each CH can simply use a HELLO message to identify the dead nodes count in its cluster in each round.

The working procedure of this step can be summarized as follows: the cloud compares the latest reconstructed data x′ with X and decides whether to re-generate depending on the error value, ε=||x−x′||, where ||·|| is Ln−norm. If it goes beyond a predefined threshold β which means that there is a change in network status, the cloud regenerates new ξ, otherwise no need to change last seed.

(D) CH rotation

CHs check the piggybacked CM–REs information to make decision on whether to continue as CHs or give up their CH roles to any other node in their respective clusters based on RE and assign these nodes as the new CHs. This step prevents WSNs from dying earlier by balancing the energy consumption. The whole process of the data compression phase can be seen in Fig. 4.

Figure 4 Flow chart of DCCS Data Compression Phase.

RSMP algorithm

In this section, we propose a new reconstruction technique called RSMP. RSMP can be utilized by the cloud to recover the sensor readings again. It is a reversible greedy algorithm in the sense that it has reversible construction, the support set can be pruned (backward step) to eliminate the unreliable elements chosen in the past (forward step). Before presenting the RSMP algorithm, we define some operations which we use in the algorithm as provided below:

(5) resid(y,x)≜y–Φx

(6) supp(x;k)≜{thesetofindicesthatcorrespondstotheklargestamplitudecomponentsofx},

(7) rand(x;k)≜{thesetofindicesthatcorrespondstothekrandomlychosencomponentsofx},

During the forward step, most of the CS reconstruction greedy algorithms used Matched Filter Detection (MF) operation Φ′y to calculate the correlation between matrix Φ columns and the Sampled Measurement Vector y. Then, Eq. (6) is used to select the set of indices that corresponds to the n largest amplitude components of Φ′y. The size of n may vary for each algorithm, for example: n = 1,.. s, and 2S in OMP (Tropp & Gilber, 2007), SP (Wei & Olgica, 2009) and COSAMP (Needell & Tropp, 2009) algorithms respectively. However, as a result of measurement noises, the MF does not usually give the indices of all correct columns. Indeed, the correct indices may not be selected because they give small correlation according to Eq. (6). To solve this drawback, RSMP proposes a random technique to the selection process in the forward step to increase the probability to find the correct column indices in each iteration. Figure 5 provides the working of RSMP algorithm.

Figure 5 RSMP Algorithm.

The proposed algorithm includes four steps: initialization, forward, backward and update as detailed below:

(A) Initialization: the proposed algorithm initializes all parameters as follows: initial approximation E0 = 0, residual r r0 = y, and estimated set T = φ.

(B) Forward: the main contribution of RSMP algorithm is in this step. Most of the MP algorithms use the n largest in-amplitude components from the MF, |n| depends on the algorithm, as a first estimation of the estimated set T. However, they depend only on the high correlation columns in MF equation without taking consideration the others which have negative effect on the reconstruction performance especially when the sparse level increases. Due to the measurements noise, the correct columns do not usually give high correlation during MF process.

Random Selection Matching Pursuit algorithm uses a simple way to improve this step. Instead of choosing the indices corresponding to largest amplitude components in the set of F only, in each iteration, RSMP selects s + q columns where q is the random selection size. RSMP firstly selects the largest s components in F (H = supp(F, s) ) to create set H and then uses Eq. (7) to select q random components from set F(R = Rand(F, q)), and creates set R to overcome the challenging case in which the correct columns do not give high correlation. Indeed, the probability to find the correct columns in both cases is increased. RSMP sets q = m/2 – s according to the fact that the CS signal recovery problem can be solved if s ≤ m/2, where S is the sparsity level (Al-Zubaidi, Ariffin & Al-Qadhi, 2018). Finally, RSMP uses the union set U = H ∪ R between set H and set R to expand the estimated set T and start the next step.

(C) Backward: we can call this step as correction step because through this step, the RSMP algorithm eliminates incorrect column indices which were incorrectly selected in the last step, i.e the technique updates the approximation set Ek = W|s by removing s column indices that have the least values in set W.

(D) Update: the samples are updated using Eq. (5) as rk = resid(y, Ek). There are two situations that terminate our algorithm: (1) the algorithm will stop when the residue set r ||rk||2 is lower than the β which is the termination parameter. The selection of β is based on the noise level; (2) If the number of iterations exceed kmax where kmax is the maximum count of iterations. At the end, Ekholds the corresponding nonzero values

Experiments

This section includes the results of simulation for analyzing the performance of our work. We divide this section into three parts: in the first part, DCCS technique is evaluated with reference to (i) network lifetime (first node die) and (ii) average energy consumption. In the second part, we analyze the RSMP reconstruction technique in comparison to OMP, COSAMP, Forward-Backward Pursuit (FBP) (Burak & Erdogan, 2013), SP (Wei & Olgica, 2009) and E-OMP algorithms (Osamy, Salim & Aziz, 2014). Finally, the dynamic re-generation of random seed step is evaluated in terms of average power consumption and reconstruction error in the third part.

Evaluation of DCCS algorithm

In this section, we describe the details of the simulations performed in MATLAB environment. The network region is having a size of 100 × 100 m, and the BS is located at the region center. The nodes count is varied from 50 to 200 nodes with an incremental factor of 50.

This section is subdivided based on two cases: Homogenous network and Heterogeneous network.

Performance Metrics: we use the following performance metrics to analyze and compare the proposed algorithm performance with baseline algorithms:

(1) Average Energy Consumption: it is given by the total energy expended by the entire nodes divided by the total number of nodes during their operations like sending, forwarding and receiving. The average energy consumed for each round can be estimated as:

(8) Eaverage=∑i=1N⁡Ei(r)r

where n denotes the nodes count and r refers to the round.

(2) Network lifetime: we measure the lifetime of the network according to the first node death.

Case 1: homogeneous network

In this case, DCCS algorithm performance is evaluated in comparison with DMC-CS (Salim & Osamy, 2015) and EDACP-CS (Khedr, 2015). We use the same energy parameters as used in Salim, Osamy & Khedr (2014). To send a message of size l-bits to a distance d, the power consumed by the radio, we use Eqs. (3) and (4) in “Network Assumptions”.

Figure 6 shows the lifetime for DCCS, EDACP-CS and DMC-CS. In EDACP-CS and DMC-CS, the death of the first node is earlier than in DCCS, and also Fig. 6 shows the potency of the DCCS algorithm in enhancing the lifetime of the network than compared to EDACP-CS and DMC-CS algorithms. The reason is that the DCCS uses a fixed count of CHs (NCH) per round, which leads to achieve better stability in energy utilization among the nodes, when compared to other algorithms. Additionally, in DCCS, the BS takes the role to select the CHs only in the first round and then the CHs change dynamically, which considerably decreases the overhead cost of computation associated with CH search when compared with others. DCCS reduces the transmitted CS measurement samples in each cluster which dynamically depends on the network status rather than using a fixed number of CS measurement samples in each round as in other algorithms.

Figure 6 Network lifetime in DCCS, DMC-CS and EDACP-CS.

Figure 7 depicts the lifetime and the count of alive nodes in the network per round for DCCS, EDACP-CS and DMC-CS. It clearly shows that the first and last node death in DCCS happen several rounds later than those of EDACP-CS and DMC-CS, which means that DCCS minimizes the energy utilization among all sensors. This is because DCCS reduces the power consumption of each node by organizing the nodes of each cluster in a chain such that each node sends and receives only from the nearest node, which is not considered by EDACP-CS algorithm. During the chain construction, DCCS rearranges all nodes in the chain when it adds a new node to the chain list to take into consideration the distances between that node and the others in the chain, rather than simply adding the closest node as the last node of the chain like DMC-CS. From Fig. 8, it is evident that DCCS succeeds to decrease the average energy consumption when compared to EDACP-CS and DMC-CS algorithms. The main reason for this is due to the dynamic re-generation of CS matrix in DCCS, which is not considered in the other algorithms.

Figure 7 Count of alive nodes as a function of number of rounds.

Figure 8 Average energy consumption in DCCS, EDACP-CS and DMC-CS.

Case 2: heterogeneous network

Here, we focus to evaluate the proposed algorithm performance in a heterogeneous network scenario. In this case, we make an assumption that the total network energy is 102 J, where the nodes are divided into advanced, intermediate and normal nodes according to their residual energy. DCCS performance is evaluated in comparison with ETSSEP (Kumar, Kant & Kumar, 2015), SEECP (Mittal, Singh & Singh Sohi, 2017) and SILEACH (Al-Zubaidi, Ariffin & Al-Qadhi, 2018) based on CS method. It’s clear that DCCS still provides good performance with reference to network lifetime in terms of first node dies enhancement in comparison with ETSSEP, SEECP and SILEACH algorithms as shown in Fig. 9. That is because the dynamic CS matrix regeneration process in DCCS gives it the ability to utilize CS matrix in an effective way to minimize the total transmitted data which leads to reduce the transmission energy expenditure. Whereas, the other algorithms use the same CS matrix in each iteration which may become inappropriate for the network after a number of iterations. The same effect can be noticed in Fig.10 where DCCS performs better than the other algorithms with reference to network lifetime in half-node death case.

Figure 9 Network lifetime (First node dies) in DCCS, ETSSEP-CS, SEECP-CS and SILEACH-CS.

Figure 10 Network lifetime (half of nodes die) in DCCS, ETSSEP-CS, SEECP-CS and SILEACH-CS.

From Fig. 11, we can conclude that DCCS succeeds to minimize the total energy expenditure in comparison with the others. That is because DCCS divides the network into various clusters and inside every cluster, it uses the proposed chain construction algorithm to arrange the CMs into a chain. In addition, DCCS uses the same proposed chain construction algorithm to organize the FN transmission to the Cloud.

Figure 11 Residual Energy in DCCS, ETSSEP-CS, SEECP-CS and SILEACH-CS.

Evaluation of RSMP algorithm

Here, we evaluate RSMP reconstruction algorithm performance in comparison with OMP, COSAMP, SP, FBP and E-OMP. Firstly, we make use of the proposed algorithm to recover the signals captured from 54 sensors placed at Intel Berkeley Research Lab. The entire experiment process is iterated for 500 times on randomly generated S sparse samples. Secondly, RSMP algorithm is applied to reconstruct computer-generated signals in which its nonzero coefficients are drawn from Uniform and Gaussian distributions. Finally, RSMP performance is measured over signal noise observations. We have adopted MATLAB environment for performing the simulations. The signal reconstruction performance is analyzed using Gaussian matrix Φ of size m × n, where n = 256 and m = 128.

Performance metrics

Random Selection Matching Pursuit algorithm performance in signal reconstruction is compared with other reconstruction algorithms with reference to Average Normalized Mean Squared Error (ANMSE), which is the average ratio ||x−x∼||2||x2||, where x and x∼represent the original and reconstructed readings respectively.

Experiments over real datasets

In this section, we use RSMP algorithm for reconstructing the signals obtained from Intel Berkeley Research lab.

Figures 12A and 12B: shows the effectiveness of RSMP algorithm in terms of reconstructing the temperature signals. RSMP achieves similar performance in reconstructing the humidity signals as shown in Figs. 12C and 12D. In Fig. 13, we illustrate the distribution of relative reconstruction error for different reconstruction algorithms. It is evident that RSMP algorithm exceeds the performance of other greedy algorithms, i.e., the COSAMP, OMP, EOMP, FBP and SP respectively.

Figure 12 Intel temperature and humidity: (A) Original temperature signal, (B) reconstructed temperature signal, (C) original humidity signal and (D) reconstructed humidity signal.

Figure 13 Reconstruction performance of six different algorithms for temperature signals.

Different coefficient distributions: In this part of simulation, Uniform and Gaussian distributions are utilized to draw the non-zero values of the sparse signal and the sparse level S ranges from 5 to 60. In Fig. 14 where the sparse signal’s non-zeros values are taken from uniform distribution, RSMP algorithm has lower ANMSE comparing to COSAMP, FBP, OMP, E-OMP and SP. Moreover, ANMSE for RSMP algorithm appear to rise only when s > 49 while it increases when s > 42, s ≥ 34, s ≥ 44, s ≥ 38 and s ≥ 41 for COSAMP, OMP, E-OMP, FBP and SP algorithms respectively as shown in Fig. 14. Figure 15 shows the results of ANMSE when the non-zero entries of sparse signal are taken from Gaussian distribution. From Fig. 15, it is evident that RSMP algorithm still gives least ANMSE result when compared to COSAMP, OMP, EOMP, FBP and SP, as s > 59, s ≥ 46, s > 34, s > 49, s > 47 and s > 45, respectively.

Figure 14 Reconstruction results over sparsity levels (Uniform Distribution).

Figure 15 Reconstruction results over sparsity level (Gaussian distribution).

Reconstruction performance over different measurement vector lengths: This part of simulation aims to test RSMP reconstruction performance when different measurement vector lengths- m are used with two different CS matrices: Gaussian and Bernoulli distribution matrices as shown in Figs. 16 and 17 respectively. To achieve this aim, sparse signals taken from Uniform distribution having length n = 120 is utilized and m values from 10 to 60 with step size of 1. From those figures, we can understand that RSMP algorithm still provides the least ANMSE values when compared to other algorithms.

Figure 16 Reconstruction results over Gaussian matrix for different lengths of M.

Figure 17 Reconstruction results over Bernoulli matrix for different lengths of M.

Reconstruction over noisy signal: In this part, we add some noise equal to 10−4 to the original Uniform as well as in Gaussian distribution signal where n = 256 and m = 128. The CS matrix Φ is drawn from the Gaussian distribution. The sparsity S levels are from 10 to 60 with step size 1.

Figures 18 and 19 depict the reconstruction errors for the noisy Uniform and Gaussian sparse signals. We can see that RSMP algorithm produces less error than COSAMP, OMP, E-OMP, FBP and SP. In summary, RSMP algorithm improves the reconstruction process and gives better performance than COSAMP, OMP, E-OMP, FBP and SP algorithms. This is because in each iteration RSMP gives the chance to the columns which do not give the largest values in MF process to be chosen.

Figure 18 Reconstruction results for noisy Uniform sparse signals.

Figure 19 Reconstruction results for noisy Gaussian sparse signals.

Evaluation of dynamic re-generation of random seed step

In this part, network area is assumed to be 100 × 100 m, having the sensor nodes count ranging from 50 to 200 nodes with an incremental factor of 50 and the BS is placed at (x = 50, y = 50).

Performance metrics

We call DCCS algorithm as DCCS-dynamic if it uses the proposed dynamic re-generation of random seed and DCCS-static otherwise. This section compares the performance of DCCS-dynamic and DCCS-static, with reference to the following: Average Energy Consumption and Average Normalized Mean Squared Reconstruction Error (ANMSE). During the reconstruction process, COSAMP (Needell & Tropp, 2009) algorithm is used to recover the data in each round.

Figure 20 shows the performance of DCCS algorithm in both dynamic (DCCS-dynamic) and static (DCCS-static) mode in terms of number of alive nodes per round. According to DCCS-dynamic scenario, the number of measurement samples transmitted in intra or inter-cluster communication decreases while the count of dead nodes are increased. Moreover, DCCS-static uses a fixed CS matrix whatever the count of alive nodes per round. On the other hand, DCCS-dynamic uses the threshold β value with reference to the best reconstruction error and then compares the reconstruction error in each round with β. If the error is larger than β, the old matrix is considered as not suitable and therefore regenerates another one.

Figure 20 Number of Alive nodes in DCCS-dynamic and DCCS-static.

Conclusions

The main objective of IoT components is to collect accurate information about any event. However, there are some challenges that hinder the way to attain this objective such as sensor battery constraints and dealing large amount of data acquisition. To solve these problems, this research introduced a new CS scheme for IoT and explained how this scheme could be utilized to compress and reduce the overall data traffic through the network. The proposed work consists of two algorithms; the first one is the DCCS algorithm which converts the network into several clusters and organizes each cluster into chain to start the CS data gathering. The second algorithm, RSMP, is used in the cloud side in order to reconstruct the original data successfully. In each round, the cloud checks the suitability of the measurement matrix to the network to decide whether to change or not. The proposed work achieved our objectives to enhance the IoT network lifetime and improved the reconstruction performance. Simulation results proved that our proposed algorithm is an effective data acquisition tool for decreasing the energy consumption in networks.

Supplemental Information

Supplemental Information 1 FBP reconstruction Algorithm.

Click here for additional data file.

Additional Information and Declarations

Competing Interests

Author Contributions

Data Availability

The authors declare that they have no competing interests.

Walid Osamy conceived and designed the experiments, performed the experiments, analyzed the data, performed the computation work, prepared figures and/or tables, authored or reviewed drafts of the paper, and approved the final draft.

Ahmed Aziz conceived and designed the experiments, performed the experiments, analyzed the data, prepared figures and/or tables, authored or reviewed drafts of the paper, and approved the final draft.

Ahmed M. Khedr conceived and designed the experiments, performed the experiments, analyzed the data, authored or reviewed drafts of the paper, and approved the final draft.

The following information was supplied regarding data availability:

MATLAB code is available as a Supplemental File.

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
