# Peer review of "Deterministic clustering based compressive sensing scheme for fog-supported heterogeneous wireless sensor networks"

_PeerJ Computer Science, doi:10.7717/peerj-cs.463_

## Round 0.1 · original submission · Major Revisions

According to the comments, reviewers expressed concerns about the presentation, quality and novelty of the work, as well as missing of some important technical explanations. Authors should carefully address all the comments with clear indication of the changes in the revised manuscript.

Reviewer 1 ·

Basic reporting

no comment

Experimental design

no comment

Validity of the findings

no comment

Additional comments

Lines 22-23, “An open question in this approach is how to integrate … for specific tasks”. What is the approach you mention ? this is not clear for readers. The interest for merging CS and rouing protocols must be more shown.
Line 59, A “.” is missing after [5].
Lines 77-78, the definition of parameters is not well done (e.g. belonging symbol)
Line 81, please check the expression “Then ||L||1 –norm minimization”.
Lines 82-83, the text alignment becomes center, fix this issue.
Line 83, Equation (2) is not well defined.
Line 84, what is considered as the most important communication? Remember the main idea.
Line 108, the “and” can be replaced by “;” or a “,”
Line 229, “The working of DCCS” can be replaced by “The functioning of DCCS”
Line 233, “The basic aim” can be replaced by “The main objective”
Line 234, “X” is a variable that gives the amount of data? please explain it
Line 237, “… depends only on the RE of the nodes …” can be replaced by “… depends only on the residual energy (RE) of nodes”
Line 241, I suggest to remember how the reference [50] find the number of CH (NCH)

• The paper is well organized and the english language is acceptable. However. There are few incomprehensible expressions;
• We suggest to put the paragraph (Lines 182 - 201) rather in the Introduction before the paragraph of the paper organization;
• According the CH selection description in lines 237-246, the CHs elected are the NCH nodes with the highest residual energy ? If so did you assume each CH can communicate with all the FN ?
• The way the Fog leader (You should explain this agent) is choosen is confused and not explain ? This is because early you said that the positions of FNs are predefined (i guess also their location) and details about the selection of fog leader is missing, or is it predefined ?
• In Algorithm 1 (Lines 266) I think “Ci” must be replaced by “ch in Ci” because Ci is the set of CH and you want each non-CH nodes check its closest CH
• Another simple way to improve Algorithm 1 is to consider CHset as the set of CHs and Ci is the i-th CH of the set.
• When designing the DCCS algorithm, you assumed the location of Fog Nodes (FN) are predetermined, during your experimentations; the position of the other nodes (normal and advanced nodes) is randomly achieved? If yes, how many tests did you performed and what were the standard deviation of the tests.
• What is the nature of the DCCS algorithm you proposed? is it a centralized or a distributed clustering approach ?
o Authors have to clarify this issue!
o The selection of the CH is it performed through a deistibuted or centralized manner ?
• I am confused during the evaluation of the RSMP, indeed, the authors use MATLAB for simulation, however, is MATLAB appropriate for simulation in cloud environment? Can you please that choice?

Annotated reviews are not available for download in order to protect the identity of reviewers who chose to remain anonymous.

Reviewer 2 ·

Basic reporting

The professional writing used throughout this paper need minor revision; as the document contains some grammatical and typing errors.
The structure of the document should be revised, as it makes it difficult for readers to follow.
Authors should add related works on data reconstruction methods.
More details are given in the attached files.

Experimental design

To tackle the data acquisition problem in large scale WSNs, the authors proposed a routing protocol based on deterministic clustering combined with the compressive sensing. Nevertheless, the performance evaluation is not consistent, hence needs major revisions.
More details are given in the attached files.

Validity of the findings

In this paper, the authors proposed a routing protocol based on clustering, a compressive sensing scheme and a data reconstruction algorithm to take energy consumption and data acquisition problems in large-scale wireless sensors networks. The document is poorly structured that it did not highlight the originality of this work. Therefore, it requires major revisions. Detailed comments provided will help the authors highlight the importance of novelty of this work.

Additional comments

To tackle the data acquisition problem in large scale WSNs, the authors proposed a routing protocol based on deterministic clustering combined with the compressive sensing.
The authors are highly advised to review the quality of the paper. Since it contains several grammatical and typing errors. The use of software such as Grammarly may help to improve the document quality.

Annotated reviews are not available for download in order to protect the identity of reviewers who chose to remain anonymous.

---

## Round 0.2 · Minor Revisions

Thanks authors to address most of the comments from reviewers. However, one of the reviewer is asking for some minor changes in your manuscript. Authors should have quick revision for the comments before this paper can be finally accepted.

Reviewer 1 ·

Basic reporting

The remarks and suggestions have been addressed.

Experimental design

Enough for publication in PeerJ Computer Science.

Validity of the findings

no comment

Additional comments

no comment

Reviewer 2 ·

Basic reporting

The authors answered most of the comments I raised. The paper has been restructured. Nevertheless, authors are strongly recommended to proofread this document; the writings may not encourage more readers.

Experimental design

no comment

Validity of the findings

no comment

Additional comments

See the attachment

Annotated reviews are not available for download in order to protect the identity of reviewers who chose to remain anonymous.

---

## Round 0.3 · accepted · Accept

Thanks authors to revise the comments based on previous round of the review. All the comments now were answered and therefore, it is recommended to publish this paper in its current form.